# Birth Traits Associated with Pre-Adulthood Disease Manifestations in Calves

**DOI:** 10.3390/ani14192844

**Published:** 2024-10-02

**Authors:** Jiayu Yang, Zhangping Yang, Zhipeng Zhang

**Affiliations:** 1College of Animal Science and Technology, Yangzhou University, Yangzhou 225009, China; mz120221551@stu.yzu.edu.cn (J.Y.); yzp@yzu.edu.cn (Z.Y.); 2Zhejiang Key Laboratory of Cow Genetic Improvement and Milk Quality Research, Wenzhou 325000, China; 3Experimental Farm, Yangzhou University, Yangzhou 225009, China

**Keywords:** calf, birth traits, health traits, animal welfare

## Abstract

**Simple Summary:**

Calf health is a critical consideration for animal welfare and farm profitability. Diarrhea and pneumonia are the two most common diseases threatening calf health. In addition to diseases, birth traits have been recognized by farms in recent years as important indicators of calf health and welfare. Given this, currently people are not only concerned about the health status of calves, but increasingly focused on the birth traits of calves derived from a reproductive perspective, including gestation length, calf weight at birth, and calving ease score. The period of calf growth and development spans from birth to six months of age. We believe that the birth traits of calves should be correlated to some extent with their health status during this growth and development period. This study assessed the correlation between calf birth traits and their disease manifestations during the growth and development period. We have found that the length of pregnancy in the dam and the season of calf birth may influence the duration of treatment for calf diarrhea and pneumonia. Additionally, we believe that calves born in winter may require increased care or higher medication dosages after illness onset. Moreover, an ideal gestation period for cows may be between 278 and 282 days, which could potentially result in lower rates of pneumonia occurrence in calves or shorter treatment durations for pneumonia.

**Abstract:**

The objective of this study was to explore the relationship between calf birth traits and their susceptibility to diseases before reaching adulthood. A total of 5253 birth traits of Chinese Holstein calves were examined, including gestation length (GL), calf weight at birth (CW), and calving ease score (CES), which ranges from 1 (easy) to 5 (very difficult). Furthermore, monthly medical records were scrutinized for pneumonia and diarrhea in these calves. The study assessed five aspects of disease manifestation in calves: age at first onset, frequency of illness, longest duration of treatment, and total duration of treatment. The link between age at onset and disease manifestation prior to adulthood was analyzed using general linear models and regression models. The GL of calves significantly correlated with the risk of pneumonia, with the risk decreasing as the GL increases. A higher CES was associated with a later onset of diarrhea in calves. Furthermore, the CES was significantly negatively correlated with the duration of diarrhea treatment in calves. These results suggest that implementing different preventive measures for calves with different birth traits and modifying treatment protocols for affected calves could enhance the productivity of dairy cows and reduce losses on farms.

## 1. Introduction

In general, the calf stage is defined as the period from birth to six months of age. Calves are the future of a dairy farm, and their early growth and development are closely related to the farm’s production performance and economic benefits [1,2]. The occurrence of syndromes during the growth of calves significantly affects the adult body size [3], lactation performance [4], and reproductive performance of dairy cows [2]. Pneumonia and diarrhea are common syndromes in calves, with the highest mortality rates occurring within the first few weeks after birth [5]. Research indicates that approximately 35% of calves experience illness during the calf stage [6]. Moreover, the risk of mortality within the first year after birth ranges from 2.1% to 14%, with the exact value varying depending on the calf’s age, population, and year of birth [7]. The occurrence of these syndromes in calves increases treatment costs and adversely affects their growth performance [8]. Calves infected with pneumonia show a significant impact on their weight gain. Heifers treated for respiratory disease continued to have lower body weights as they grew to 400 days of age compared to calves that did not develop respiratory syndrome [9]. Calves that experience diarrhea within the first few weeks after birth have higher mortality and culling rates before their first calving [10]. Cattle that experienced diarrhea during the calf period tend to be older at their first calving compared to those that did not suffer from diarrhea. Additionally, the survival rates of the first and second calves produced by these cattle are lower [11]. These bacterial or viral infections may lead to systemic infections in calves, such as bovine viral meningitis, septicemia, or other complications. Additionally, the use of antimicrobials for treatment may result in the affected cattle developing antibiotic resistance, thereby exacerbating the challenge of effectively treating future infections [12]. Therefore, developing technologies or methods for early screening of disease risk in calves, followed by targeted care, is of great significance for farm production operations.

Previous research has indicated that there are many factors influencing the health traits of calves, with the primary ones including genetics, nutrition, environment, and birth traits. Researchers have investigated the heritability of susceptibility to diarrhea and respiratory syndromes in dual-purpose calves in Austria, finding it to be 0.027 for diarrhea and 0.039 for respiratory diseases [13]. It is worth noting that calf birth traits, as significant indicators of calf welfare and critical components of reproductive performance, have received increasing attention in recent years. These traits include factors such as gestation length, birth weight, and calving ease score [14]. In previous studies, we found that the characteristics of calves at birth are directly related to their subsequent growth and development [15]. Calving ease scores are associated with decreased survival rates of both cows and calves, as well as reduced longevity, productivity, and reproductive performance in cows [16]. Dystocia not only causes harm to cows but also increases the economic burden on farms. It leads to higher mortality rates in newborn calves, reduced milk production, and an overall decline in the health status of cows [17]. An increase in gestation length and birth weight lead to higher rates of stillbirth and dystocia in calves [18]. The direct relationship between birth traits and syndromes has not been explicitly discussed in the literature. Further research is needed to explore the specific link between calf birth traits and morbidity. Understanding these relationships can help formulate preventive measures and management strategies to improve calf health and reduce syndrome incidence in the early stages of their lives.

The aim of this study was to investigate the association between calf birth traits and common diseases, serving as a predictive method based on calf birth traits for early warning of potential health issues. By identifying potential health risks early, farm staff can take appropriate preventive measures at the time of calf birth, thereby further aiding in the care and treatment of syndromes in calves.

## 2. Materials and Methods

### 2.1. Data Sources

The data for this dairy cattle study were obtained from a standardized management farm in Jiangsu Province, China. The farm is a non-experimental facility. The farm operates a double-row barn system, allowing for free-range grazing of the dairy cows. They are fed a total mixed ration (TMR) three times daily, and standardized milking and calving practices are implemented. The farm also adheres to strict feeding management protocols, with an annual cow population exceeding 6000 heads. The average milk production of cows was similar at 4900 ± 49 L. This study collected records of 2978 cases of diarrhea and pneumonia in Holstein calves from birth to 6 months of age, from a total of 5253 heifer calves at a dairy farm between 2020 and 2022. The average calf is born at a parity of 2.31 ± 0.12. Calves remained with their dams for 48 to 72 h after birth, after which they were housed individually until 3 months of age. The calves were fed 4 L of unpasteurized fresh milk at a temperature of 37 °C daily and provided with access to calf starter feed for ad libitum consumption. After weaning at around 90 days, the calves were grouped into two stages based on age: 3–4 months and 5–6 months. We visited once a week to collect data on documented diseases. Additionally, on a monthly basis, we collected the birth traits of calves born during that month, including calving ease score, gestation length, and birth weight (Table 1). The birth traits in our study included calving ease score from 1 (easy) to 5 (very difficult) [19], gestation period, and calf birth weight. All measurements were taken by veterinary technicians. These indicators have been detailed in previous publications [14].

All measurements were taken by veterinary technicians. These indicators have been detailed in previous publications [14]. The age at onset of illness in calves was calculated based on the basic data derived from the birth dates. The syndrome records included the animal’s identification number, the onset time of the illness, the nature of the syndrome, and the duration of treatment. Diarrhea and pneumonia in calves were the primary health concerns documented in the veterinary records obtained from the farm. During this period, in addition to these two conditions, there were 45 other syndrome records including trauma, conjunctivitis, internal bleeding, etc. The occurrence of these syndromes was more than one month apart from the onset of diarrhea and pneumonia in the calves, and therefore they did not have a direct impact on this study. Records showing the simultaneous occurrence of pneumonia and diarrhea in 616 calves were deleted from the dataset. Because there were multiple possible causes of calf deaths, the ranch did not autopsy the calves to confirm the diagnosis, so we removed the dead calves from the counting process. Additionally, any inaccurate or incomplete information was also removed. The farm veterinarians identified the calves suffering from pneumonia and diarrhea. The early clinical signs of pneumonia included fever, coughing, and the presence of sticky nasal discharge, with body temperatures reaching up to 41 °C. In severe cases of pneumonia, the calves exhibited anorexia, lethargy, rapid increases in body temperature, and rapid breathing [20]. The initial clinical signs of diarrhea in calves include dehydration, decreased appetite, difficulty standing, and in severe cases, ataxia, acidosis, and irregular heart rate.

The diagnosis of individual syndromes was confirmed, with the incidence of each syndrome recorded as “1” and the absence of syndrome recorded as “0”. Multiple incidences of the same individual are recorded only once. If the interval between syndrome occurrences in a calf was ≤21 days, it was considered a recurrent syndrome. Data on calves with recurrent disease have been removed. The treatment duration refers to the time from the initial onset to full recovery. The longest recorded treatment duration was the longest treatment time in cases where calves had multiple episodes of diarrhea or pneumonia. The total duration of illness was the sum of all specific syndromes occurring in the calf.

### 2.2. Statistical Analyses

Data were analyzed and processed using SPSS statistical software (SPSS 26.0, IBM, Ehningen, Germany). This study primarily focused on the association between calf birth traits and the risk of syndrome occurrence. To facilitate analysis, all variables were converted into categorical variables. Calf birth weight and gestation period were divided into high, medium, and low levels based on quartiles (Table 2), using the upper and lower quartiles. The parity was divided into three groups: 1st parity, 2nd parity, and 3rd parity. According to the seasonal climate characteristics of Jiangsu Province, the divisions are as follows: spring (March to May), summer (June to August), autumn (September to October), and winter (December to February) of the following year.

The relationship between calving season, parity, calf birth weight, gestation period, calving difficulty, and the occurrence of pneumonia and diarrhea in cattle was analyzed using logistic regression models. To avoid collinearity, we performed statistical analyses separately for *GL_m_* and *CW_n_*. The models are as follows:$$Logit\;\left(p\right)=\ln(\frac{p}{1-p})=\beta_{0}+S_{i}+P_{j}+A_{k}+\beta_{1}*{CES}_{l}+\beta_{2}*{GL}_{m}+e_{ijklm,}$$
$$Logit\;\left(p\right)=ln(\frac{p}{1-p})=\beta_{0}+S_{i}+P_{j}+A_{k}+\beta_{1}*{CES}_{l}+\beta_{3}*{CW}_{n}+e_{ijkln,}$$
where p represents whether the calf had pneumonia or diarrhea, $\beta_{0}$ is the constant term, $S_{i}$ is the ith fixed effect of year-season, $P_{j}$ is the jth fixed effect of parity, $A_{k}$ is the kth fixed effect of age at first calving, and ${CES}_{l}$ is the th fixed regression effect of calving ease score. ${GL}_{m}$ is the mth fixed regression effect of gestation length, ${CW}_{n}$ is the nth fixed regression effect of calf birth weight, and $e_{ijklm},e_{ijkln}$ represents the random residual. The $\beta_{1}-\beta_{3}$ are regression coefficients for each effect, respectively.

The odds ratio (OR) was used to reflect the strength of the association between each characteristic and the occurrence of pneumonia or diarrhea. It can be interpreted as the probability of pneumonia or diarrhea for a particular characteristic relative to the probability of pneumonia or diarrhea for other different characteristics. Confidence intervals indicate the extent to which the true value of a parameter is likely to be close to the measured result. The confidence interval provided a measure of the reliability of the parameter estimates.

Multiple comparisons were performed using Duncan’s multiple range test. The specific effects of season, parity, calf birth weight, gestation period, and calving ease score on the longest treatment duration, total treatment duration, incidence frequency, and age at onset of pneumonia or diarrhea were analyzed. The model is as follows:$$y_{ijklmn}=\mu_{0}+S_{i}+P_{j}+A_{k}+{CES}_{l}+GL_{m}+{CW}_{n}+e_{ijklmn},$$
where $y_{ijklmn}$ represents the observed values of the calf’s longest treatment time, total duration of illness, illness frequency, and age at onset; $\mu_{0}$ denotes the overall mean; $S_{i}$ is the *i*th fixed effect of farm, year, and season; $P_{j}$ jth is the fixed effect of parity; $A_{k}$ is the kth fixed effect of age at first calving; and ${CES}_{l}$ is the lth fixed effect of calving ease score. ${GL}_{m}$ is the mth fixed effect of gestation length, ${CW}_{n}$ is the nth fixed effect of calf birth weight, and $e_{ijklmn}$ represents the random residual. *p* < 0.05 indicates significance, while *p* < 0.01 indicates high significance. 

Spearman correlation coefficients were used to analyze the relationships between the longest treatment duration and total treatment duration, incidence frequency, and age at onset, as well as the factors of season, parity, calf birth weight, gestation length, and calving ease score. The value of Spearman’s correlation coefficient is always between −1.0 and 1.0, variables close to 0 are made uncorrelated, and close to 1 or −1 are said to have a strong correlation. When |r| ≥ 0.7, the two variables are considered to be strongly correlated; when 0.4 ≤ |r| < 0.7, they are considered to be moderately correlated; when 0.2 ≤ |r| < 0.4, they are considered to be weakly correlated; and when |r| < 0.2, they are considered to be very weakly correlated [21].

## 3. Results

### 3.1. Association between Calf Birth Traits and the Occurrence of Pneumonia and Diarrhea

Table 3 shows the prevalence of pneumonia and diarrheal diseases. Table 4 shows the logistic regression results indicating that the incidence of pneumonia in calves is associated with the length of gestation. As the gestation period increases, the risk of pneumonia decreases. No factors were found to be associated with the incidence of diarrhea in calves (*p* > 0.05). Calves from the high gestation group had a 0.78 times lower risk of pneumonia compared to those from the low gestation group.

### 3.2. Effects of Calf Birth Traits on the Risk of Pneumonia and Diarrhea in Chinese Holstein Calves

As shown in Table 5, the season of birth and gestation period significantly affected the incidence frequency of pneumonia (*p* < 0.05). Calves born after a shorter gestation period had longer treatment durations, higher incidence, and younger ages at first onset compared to those with other gestation periods. The incidence frequency of pneumonia was significantly higher for calves born in winter compared to those born in summer (*p* < 0.05). Table 6 shows that calves born in autumn had significantly shorter durations for the longest treatment and total treatment for diarrhea compared to those born in spring (*p* < 0.01). Additionally, the age at onset of diarrhea was significantly earlier for calves born in autumn compared to those born in spring (*p* < 0.01). Calves with a calving ease score of three had a significantly later age at onset of diarrhea compared to those with a calving ease score of one (*p* < 0.05). Data in Table 5 and Table 6 are presented as least squares means ± standard error.

### 3.3. Correlation between Calf Birth Traits and Specific Syndrome Manifestations of Pneumonia or Diarrhea in Calves after Onset

We examined the relationship between calf birth traits and pneumonia using Spearman’s correlation analysis with the goal of determining whether these birth traits contribute to the treatment of pneumonia and whether they help provide better protection for calves. As shown in Figure 1 of the data matrix, we found a negative and weak correlation between the gestation period and the longest treatment time for pneumonia. Gestation length was also negatively and weakly correlated with total treatment time. In addition, parity was negatively and weakly correlated with longest treatment time and total treatment time, respectively, while season was positively and weakly correlated with age at onset.

As shown in Figure 2 of the data matrix, the results indicated that no correlation was found between calf diarrhea and calf birth traits, but the distribution of age at onset was negatively and weakly correlated with the longest duration of diarrhea and frequency of illness.

## 4. Discussion

Diarrhea and pneumonia during the calf period can lead to an increase in premature culling and a delay in conception and first calving age, resulting in economic losses and increasing the environmental impact of the cattle industry. Our study aimed to explore the correlation between calf birth traits and the incidence of diarrhea and pneumonia. By understanding these associations, we aimed to enable discussion on the potential to provide recommendations for early calf care to prevent syndrome-related losses and to offer improved treatment strategies for calves with different birth traits, ultimately reducing costs for farms [2]. In this study, we observed that calves born to cows with shorter gestation periods are more susceptible to pneumonia. Calves born after a gestation period longer than 278 days were less likely to develop pneumonia during their growth. Previous research has shown that the critical period for lung development in calves is between 240 and 260 days of gestation. Premature termination of pregnancy can result in incomplete lung development in calves, leading to respiratory distress syndrome in newborn calves. In our study, all the calves were born with a gestation period exceeding 260 days, indicating that their lungs should have been fully developed. None of the newborn calves experienced respiratory distress syndrome. However, during the growth and development of the calves, we observed that calves born from cows with shorter gestation periods were more prone to developing pneumonia, a respiratory tract syndrome, compared to calves born from cows with longer gestation periods. This raises the question of whether there is a correlation between lung development and the occurrence of pneumonia in calves, which merits further investigation [22]. Our correlation analysis revealed that the birth weight of calves is significantly positively correlated with both parity and gestation period, and gestation length was negatively correlated with the longest treatment time for pneumonia in calves. In our previous studies, we also found that stillbirth and dystocia are related to the birth weight of calves [14]. This suggests that extending a cow’s gestation length results in greater calf birth weights, as well as shorter pneumonia treatment times for calves with long gestation lengths. However, extending gestation length indefinitely is clearly risky for the farm. Researchers have shown that inducing labor in cows with a gestation period exceeding 282 days could increase the chances of these cows becoming pregnant again and reduce the risk of calf mortality [23]. However, other researchers have shown that for induced abortions, monotherapy with dexamethasone or prostaglandins in pregnant heifers leads to a high incidence of retained fetal membranes, which adversely affects postpartum fertility [24] and milk production in heifers [25]. Therefore, we recommend that the optimal gestation period for dairy cows on farms should be between 278 and 282 days. For cows with gestations greater than 282 days, the long-term consequences to the cow, including increased risk of involuntary culling, should be weighed against the farm cost savings of recommending induction to ensure lactation while reducing the time spent treating calves with pneumonia. Calves born to cows with a gestation period shorter than 278 days require further attention from veterinarians and livestock handlers. These calves will need to have their feeding environment improved as well as their nutrition supplemented after their birth, and may need increased doses of medications or additional supportive medications as well to shorten the duration of the disease when they become ill. In the regression analysis of pneumonia, although seasonality did not have a significant effect on the incidence of pneumonia, the risk of developing pneumonia in winter was lower compared to summer, relative to spring. This differs from the results of other studies, where during winter, farms often reduce ventilation and increase calf stocking density to maintain barn temperature, which leads to an increased incidence of calf pneumonia [26,27]. The results of our study may be due to the selected farm having effective winter management practices, including good control of stocking density and proper ventilation while maintaining barn temperature. Moreover, due to the limited sample size of calves with pneumonia, further research with increased sample sizes and data from various farms is needed to analyze the risk of pneumonia in calves born in different seasons in this region.

In calves suffering from diarrhea, we found that those born in winter and spring have longer maximum treatment durations, overall longer treatment durations, and higher syndrome incidence rates compared to calves born [28]. This is consistent with the findings of Gutzwiller, who found spring-born calves developed acute diarrhea within 10 days of birth, while summer and fall-born calves had no cases of diarrhea within 10 days of birth [29,30].

This could be attributed to temperature fluctuations between winter and spring, which may make calves more susceptible to diarrhea. This may be due to the fact that pathogenic bacteria and parasites that cause diarrhea proliferate easily during the gradually warming season [31]. These fluctuations can create an unstable environment that can compromise the immune system of calves, making them more prone to infections that cause diarrhea [32]. The phenomenon could also be attributed to the prevalence of specific pathogens during certain seasons. Researchers have also found this phenomenon in their experiments, where calves born in winter have lower levels of viral antibodies [33]. In fact, this can help explain the observed results in our study, we hypothesize that where calves born in winter and spring exhibited lower levels of viral antibodies, leading to increased susceptibility to viral infections and more frequent episodes of diarrhea with longer treatment durations. In future management practices, it is important for farms to develop specific treatment protocols, medications, better colostrum management, and improved environmental conditions, and to avoid cleaning in closed barns to facilitate the recovery of calves born in winter and spring when they experience diarrhea.

In previous studies, it was hypothesized that a higher calving ease score in cows is associated with an increased likelihood of dystocia, which may potentially result in trauma to the calf and have implications for its physiological health and vigor. The dystocia can lead to physical injuries such as bruising or fractures, potentially impacting the overall health and developmental capacity of the calf. During a difficult birthing process, the calf may experience stress, affecting its energy levels, immune system, and overall adaptability [34]. However, in our study, we found that calves were less likely to experience diarrhea and had shorter treatment times when the calving ease score was three. Based on our previous research findings, we speculate that this may be due to the increase in calving ease score associated with a larger fetal size. A score of 0 on the calving ease score indicates easy calving without intervention, while a score of 4–5 indicates very difficult calving that needs veterinary help [19]. However, in this particular experiment, a calving ease score of three did not reach the true definition of dystocia. As fetal size increases, calves also tend to be healthier. Previous research has indicated that calves with a body condition score (BCS) of two have lower incidence rates of diarrhea and mortality compared to calves with a BCS of one [35]. In this experiment, we did not measure the BCS of the calves. Further observation and research are needed to investigate the impact of BCS on the incidence of illness in the growth and development of the birth traits of the calves. In multiple comparisons, we found that the longest treatment time was longest for calves when they were first born and shortest for second born. This phenomenon may be attributed to the higher concentration of IgG in colostrum produced by cows in their third parity compared to the IgG concentration in colostrum from first and second-parity cows [36]. Immunoglobulins play a role in passive immunity by providing antimicrobial properties and resistance against various microorganisms [37]. The lower levels of immunoglobulins in colostrum may explain why calves born from second and third-parity cows have a lower frequency of diarrhea. This may also be related to the research findings of Duncan, who pointed out that cows experience reduced nutrient intake during their first pregnancy, leading to increased energy mobilization and stress metabolism in calves postpartum [38]. Additionally, this could be due to less-than-ideal management of newborn calves, resulting in longer treatment times compared to subsequent pregnancies. Increasing the nutritional value and quality for first-parity cows on the farm [39] and providing additional colostrum feeding for calves [40] can enhance the calves’ immune system, thereby reducing the economic losses caused by calf diarrhea.

Our study found that the birth traits of calves may affect the incidence of diarrhea and pneumonia, as well as the duration of treatment. We therefore hypothesize that providing early care and tailored management practices for calves with different birth traits could be beneficial in reducing costs and improving animal welfare on the farm. 

## 5. Conclusions

This study correlated the relationship between calf birth traits and the incidence features of diarrhea and pneumonia (including treatment duration, incidence frequency, and age at first onset) in calves. The results indicate that calves born in spring and winter require longer treatment for pneumonia and diarrhea. There is a significant negative correlation between the gestation period of cows and the treatment duration for calf pneumonia. Additionally, the calving ease score is significantly negatively correlated with the incidence frequency and treatment duration of calf diarrhea. These findings suggest that in order to reduce the occurrence of calf syndromes and treatment duration, farms should consider controlling the gestation period of cows between 278 and 282 days. In cases where calf birth season, cow gestation period, or calving ease score are unfavorable, it may be necessary to adjust medication, increase dosages, and improve farm management practices. Reducing calf stocking density, ensuring proper ventilation during winter while maintaining appropriate temperatures, and ensuring a clean rearing environment can be beneficial measures to shorten treatment duration and reduce farm losses.

## Figures and Tables

**Figure 1 animals-14-02844-f001:**
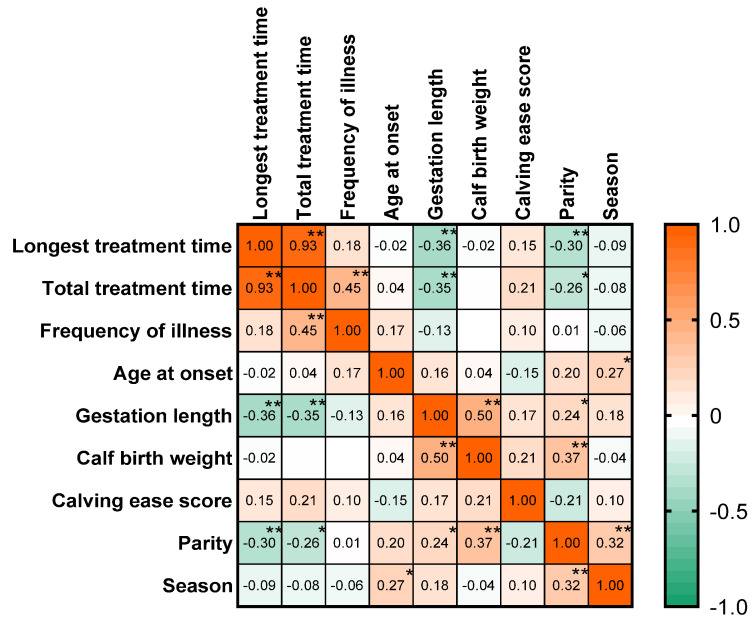
Association between manifestations of specific syndromes of calf pneumonia and birth traits after calving morbidity. Spearman correlation matrix used to compare calf birth traits and syndrome treatment outcomes for pneumonia in calves (*n* = 535). The numbers in the lower left of the matrix represent the correlation values, while the superscripts in the upper right represent the *p*-values (* *p* < 0.05, ** *p* < 0.01). The closer the color is to red, the stronger the positive correlation, and the closer the color is to green, the stronger the negative correlation.

**Figure 2 animals-14-02844-f002:**
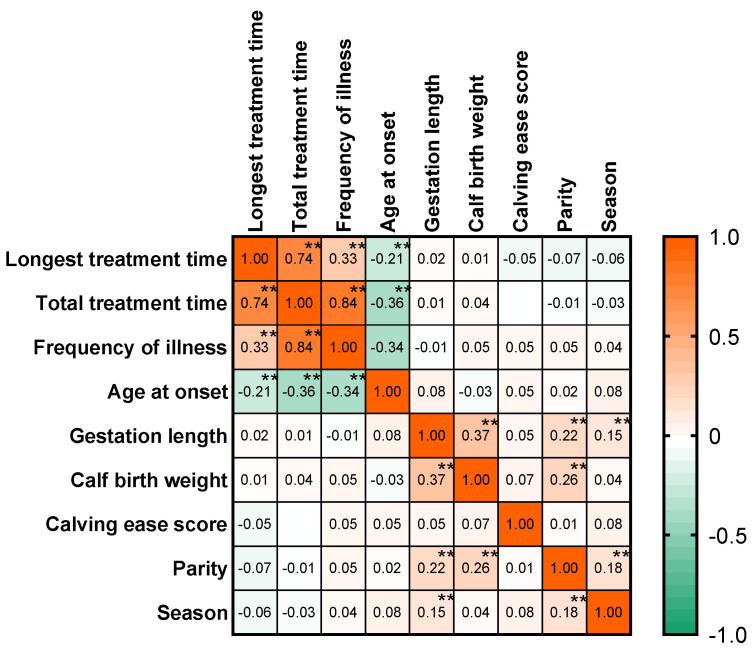
Association between manifestations of specific syndromes of calf diarrhea and birth traits after calving morbidity. Spearman correlation matrix used to compare calf birth traits and syndrome treatment outcomes for diarrhea in calves (*n* = 3941). The numbers in the lower left of the matrix represent the correlation values, while the superscripts in the upper right represent the *p*-values (** *p* < 0.01). The closer the color is to red, the stronger the positive correlation, and the closer the color is to green, the stronger the negative correlation.

**Table 1 animals-14-02844-t001:** Descriptive statistics of calf birth traits.

	Sample	Mean	SD	CV (%)	Upper Quartile	Lower Quartile
Birth Weight (kg)	5253	39.36	4.42	11.23	42	36
Gestation length (day)	5253	274.45	4.80	1.74	278	273
Calving ease score	5253	1.48	0.48	32.43		

**Table 2 animals-14-02844-t002:** Description of the quartiles of birth traits in calves.

	High	N	Medium	N	Low	N	Means
Birth Weight (kg)	42–55	1308	36–41	2616	30–35	1308	39.36
Gestation length (day)	278–284	1308	273–277	2616	261–272	1308	274.45

**Table 3 animals-14-02844-t003:** Number of calves in the breeding herd and number of calves affected by pneumonia or diarrhea and incidence of disease ^1^.

Type of Disease	Number of Calves	Total Number of Cattle	Disease Rate
Pneumonia	535	5232	10.23%
Diarrhea	3941	5232	75.32%

^1^ Multiple incidences of the same individual are recorded only once.

**Table 4 animals-14-02844-t004:** Birth traits of calves with effects on pneumonia or diarrhea ^1^.

Type of Syndrome	Factor	Level	*p* Value	OR Value	95% Confidence Interval	
					Lower Limit	Upper Limit
Pneumonia	Season	Summer vs. Spring	0.26	1.56	0.726	3.33
		Autumn vs. Spring	0.40	0.71	0.32	1.58
		Winter vs. Spring	0.11	0.44	0.16	1.22
	Parity	2 vs. 1	0.96	0.98	0.46	2.12
		3 vs. 1	0.18	1.60	0.81	3.16
	Gestation length	Medium vs. Low	0.17	0.64	0.34	1.20
		High vs. Low	0.04	0.43	0.18	0.99
	Calf birth weight	Medium vs. Low	0.52	0.81	0.43	1.54
		High vs. Low	0.73	0.86	0.78	1.99
	Calving ease score	2 vs. 1	0.19	7.41	0.38	145.34
		3 vs. 1	0.42	1.30	0.68	2.49
Diarrhea	Season	Summer vs. Spring	0.29	0.54	0.17	1.71
		Autumn vs. Spring	0.99	1.00	0.29	3.39
		Winter vs. Spring	0.91	0.93	0.24	3.59
	Parity	2 vs. 1	0.33	1.91	0.52	6.99
		3 vs. 1	0.19	0.55	0.23	1.34
	Gestation length	Medium vs. Low	0.95	0.97	0.39	2.42
		High vs. Low	0.77	1.12	0.37	3.92
	Calf birth weight	Medium vs. Low	0.28	1.63	0.68	3.90
		High vs. Low	0.42	1.58	0.52	4.82
	Calving ease score	3 vs. 1	0.17	0.56	0.24	1.28

^1^ For pneumonia and diarrhea, the baseline category for season was spring, for parity it was 1 year, for gestation period and calf birth weight it was the lower-middle quartile, and for calving difficulty it was a score of 1.

**Table 5 animals-14-02844-t005:** Mean values of specific syndromes of pneumonia in calves for each birth trait ^1^.

Factor	Group	Longest Treatment Time (Days)	Total Treatment Time (Days)	Frequency of Illness (Rate)	Age at Onset (Days)
Season	Spring	2.91 ± 1.76	3.45 ± 2.25	1.36 ± 0.51 ^ab^	4.45 ± 6.54 ^b^
	Summer	4.24 ± 1.56	4.53 ± 2.28	1.09 ± 0.29 ^b^	12.32 ± 32.25 ^b^
	Autumn	3.19 ± 1.44	3.62 ± 2.40	1.14 ± 0.47 ^ab^	43.43 ± 46.90 ^ab^
	Winter	3.17 ± 1.60	4.33 ± 3.07	1.50 ± 0.83 ^a^	29.17 ± 23.12 ^a^
F		1.69	0.37	2.82 *	2.11
Parity	1	4.00 ± 1.68	4.46 ± 2.49	1.19 ± 0.46	12.19 ± 26.40 ^b^
	2	3.46 ± 0.97	3.62 ± 1.04	1.08 ± 0.28	40.54 ± 52.13 ^a^
	3	3.14 ± 1.75	3.73 ± 2.71	1.23 ± 0.53	26.23 ± 40.67 ^ab^
F		0.87	0.78	0.56	1.25
Gestation length	Low	4.26 ± 1.71 ^a^	5.00 ± 2.64 ^a^	1.30 ± 0.55 ^a^	16.61 ± 34.12 ^b^
	Medium	3.36 ± 1.66 ^ab^	3.78 ± 2.40 ^ab^	1.17 ± 0.44 ^ab^	16.72 ± 35.04 ^b^
	High	3.31 ± 1.10 ^b^	3.31 ± 1.10 ^b^	1.00 ± 0.00 ^b^	43.92 ± 45.09 ^a^
F		3.03 *	3.89 *	3.20 *	1.67
Calf birth weight	Low	3.65 ± 1.72	4.05 ± 2.52	1.15 ± 0.49	19.40 ± 39.45
	Medium	3.54 ± 1.54	3.86 ± 1.96	1.16 ± 0.37	23.46 ± 40.10
	High	3.87 ± 1.80	4.67 ± 3.11	1.27 ± 0.59	19.93 ± 30.65
F		1.10	1.60	1.58	0.30
Calving ease score	1	3.56 ± 1.67	3.90 ± 2.36	1.13 ± 0.35	26.31 ± 42.87
	2	2.00 ± 1.50	2.00 ± 1.00	1.00 ± 0.50	31.00 ± 10.5
	3	3.95 ± 1.51	4.68 ± 2.40	1.32 ± 0.67	8.21 ± 11.82
F		0.53	0.52	1.06	1.68

^1^ *. *p* < 0.05 F represents the variation between sample means. Figures in the table are means ± standard deviation. As F becomes larger, the more variation there is between the sample means relative to the within-sample variation. Data in the same column labeled with the same letter indicate non-significant differences (*p* > 0.05), different lowercase letters indicate significant differences (*p* < 0.05).

**Table 6 animals-14-02844-t006:** Mean values of specific syndromes of diarrhea in calves for each birth trait ^1^.

Factor	Group	Longest Treatment Time	Total Treatment Time	Frequency of Illness	Age at Onset
Season	Spring	3.31 ± 1.79 ^A^	4.74 ± 2.960 ^A^	1.66 ± 0.83 ^AB^	23.43 ± 32.90
	Summer	2.77 ± 0.95 ^BC^	3.79 ± 1.90 ^B^	1.49 ± 0.66 ^BC^	26.76 ± 21.76
	Autumn	2.51 ± 0.72 ^C^	3.30 ± 1.67 ^B^	1.38 ± 0.61 ^C^	26.27 ± 10.24
	Winter	3.17 ± 0.65 ^AB^	4.33 ± 1.26 ^A^	1.50 ± 0.34 ^A^	28.17 ± 19.75
F		8.10 **	9.70 **	6.40 **	1.01
Parity	1	2.93 ± 1.38 ^a^	3.98 ± 2.34	1.50 ± 0.69 ^ab^	25 ± 23.36
	2	2.58 ± 0.79 ^b^	3.56 ± 1.79	1.46 ± 0.74 ^b^	29.11 ± 26.20
	3	2.69 ± 0.99 ^ab^	4.08 ± 5.56	1.66 ± 0.87 ^a^	22.72 ± 20.96
F		2.44 *	1.55	2.24	2.20
Gestation length	Low	2.78 ± 1.25 ^a^	3.96 ± 2.35	1.56 ± 0.76	22.19 ± 22.59
	Medium	2.76 ± 1.17 ^b^	3.87 ± 2.35	1.54 ± 0.79	27.03 ± 25.64
	High	2.89 ± 1.16 ^b^	3.98 ± 2.19	1.51 ± 0.65	24.44 ± 19.11
F		1.21	0.18	0.26	1.78
Calf birth weight	Low	2.82 ± 1.39	4.03 ± 2.76	1.55 ± 0.84	25.89 ± 22.90
	Medium	2.85 ± 1.19	3.94 ± 2.23	1.52 ± 0.72	24.33 ± 22.51
	High	2.66 ± 0.94	3.77 ± 2.00	1.56 ± 0.72	26.59 ± 25.83
F		3.43 *	1.73	0.15	0.53
Calving ease score	1	3.04 ± 1.34	4.32 ± 2.55	1.61 ± 0.81	24.11 ± 21.58 ^b^
	3	2.72 ± 1.12	3.80 ± 2.22	1.51 ± 0.73	29.28 ± 28.65 ^a^
F		1.60	1.16	0.46	2.38 *

^1^ *. *p* < 0.05; **. *p* < 0.01. F represents the variation between sample means. Figures in the table are means ± standard deviation. As F becomes larger, the more variation there is between the sample means relative to the within-sample variation. Data in the same column labeled with the same letter indicate non-significant differences (*p* > 0.05), different lowercase letters indicate significant differences (*p* < 0.05), and different uppercase letters indicate highly significant differences (*p* < 0.01).

## Data Availability

Data are contained within the article.

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
