# Peer review of "Birth Traits Associated with Pre-Adulthood Disease Manifestations in Calves"

_animals, 2024, doi:10.3390/ani14192844_

Round 1

Reviewer 1 Report

Comments and Suggestions for Authors

This is a well written and interesting paper for the most part, but .

I have some minor concerns around the introduction, where very general  statements are made that don't apply to all circumstances.  for example, 

49-57:  These statements are not referenced, or are based on a single farm.  It just isn't true that 29% of calves develop diarrhoea in most situations.

Some discussion of the reliability of the records would be useful.

Lastly, in the regressions, there is a big focus on statistical significance, with - in my view - insufficient regard to the biological importance.  For example, the odds ratio of pneumonia in Summer is 1.55 and in winter is 0.44 compared with Spring.  If you have one million animals in your study and had the same results, but they were statistically significant, this would be a major finding.  The authors do not state the proportion of animals with pneumonia, so it's difficult to interpret.

The results section should have a table showing the prevalence of the various diseases - as this allows the reader to understand the relative importance of the diseases and to understand the extent to which the paper may relate to their own individual circumstances.

With these improvements I believe it would be a good and useful addition to the literature.

Comments on the Quality of English Language

The use of English is appropriate

Reviewer 2 Report

Comments and Suggestions for Authors

Dear authors

This manuscript has a relative relevance, and hopefully will open a field of research.  I sincerely hope my comments are taken as a constructive criticism only.

Except minor modifications in the first half or so of the manuscript, it is OK.  However, statistical analysis part and discussion need a low of work to make it to a publishable paper.

I have included my suggestions for corrections in th attached file.

Comments on the Quality of English Language

I have included all my suggestions in the attached file.

Round 2

Reviewer 2 Report

Comments and Suggestions for Authors

Thanks for addressing my queries.  I have few minor comments on this occasion (in the attached PDF)

Comments on the Quality of English Language

Few minor queries (in the attached PDF)
